# Scutellarin combined with lidocaine exerts antineoplastic effect in human glioma associated with repression of epidermal growth factor receptor signaling

Xiu-Ying He[1], Yui-Si Yang[2], Yue-Xiang Zheng[2], Qing-Jie Xia[1], Hong-Zhou Yu[1], Xiao-Ming Zhao[3]*, Ting-Hua Wang[1,4]*

1 Department of Anesthesiology, Institute of Neurological Disease, West China Hospital, Sichuan University, Chengdu, China, 2 School of Integrated Traditional Chinese and Western medicine, Southwest Medical University, Luzhou, China, 3 Department of Basic Medicine, Medical School, Kunming University of Science and Technology, Kunming, China, 4 Laboratory Zoology Department, Institute of Neuroscience, Kunming Medical University, Kunming, China

* wangth_email@163.com (T-HW); sunboyzxm@126.com (X-MZ)

## Abstract

### Purpose

Glioma is the most common primary intracranial tumors. Although great achievements have been made in the treatment, the efficacy is still unsatisfactory, which imposes a hefty burden on patients and society. Therefore, the exploration of new and effective anti-glioma drugs is urgent.

### Methods

Human glioma cell lines U251 and LN229 were included in the study. Cell proliferation was detected by cell counting kit-8 (CCK8), plate clone formation assay, EdU incorporation assay and xCELLigence real-time cell analyzer. Cell apoptosis was evaluated by TUNEL assay and flow cytometry. Then, transwell assay was used for assessing the migration. Moreover, tumor xenograft model was established to examine the effect of scutellarin (SCU) and lidocaine on the growth of glioma *in vivo*. Lastly, western blot was performed to detect the protein level of epidermal growth factor receptor (EGFR).

### Results

In present study, we found that SCU and lidocaine suppressed the proliferation and migration, and induced the apoptosis of human glioma cell lines, including U251 and LN229 cells, in a dose-dependent manner *in vitro*. Moreover, the combination of SCU and lidocaine further restrained the proliferation and migration ability of U251 and LN229 cells, while induced their apoptosis *in vitro*. Additionally, SCU and lidocaine also inhibited the growth of glioma *in vivo*, and the effect of the combination was better. Above all, the toxicity of SCU and its combination with lidocaine was low to normal astrocytes and neurons. Mechanistically, the effect

**Funding:** This work was supported by Natural Science Foundation of Sichuan Province of China (Grant No. 2023NSFSC1567 to XYH), Sichuan University Innovation Research Project (Grant No. 2023SCUH0033 to XYH), Research fund projects of Yunnan Education Department (Grant No. 2020J0066 to XMZ), and Wang Wenyuan Expert Workstation Project of Yunnan Provincial Science and Technology Talent and Platform (Grant No. 202205AF150013 to THW). These funders had no role in study design, data collection and analysis, decision to publish, or preparation of the manuscript.

of SCU and its combination with lidocaine on glioma cells was partially associated with the repression of EGFR signaling.

## Conclusions

Scutellarin and lidocaine exerted a synergistic effect on suppressing the proliferation and migration and inducing the apoptosis of glioma cells, which was partly associated with the repression of EGFR signaling.

## Introduction

Glioma is one of the most common primary intracranial tumors, which is formed by a special type of brain cells (astrocytes, oligodendrocytes, ependymal cells, etc) [1–4]. The symptoms and signs caused by glioma mainly depend on the space occupying effect and the function of the affected brain regions [5–7]. In most cases, the exact etiology is unknown. More importantly, there is currently no cure for glioma. Treatments are palliative and consist of surgery, radiotherapy and chemotherapy [5–7]. However, the operation is traumatic and radiotherapy is often insensitive. Therefore, the exploration of new and effective anti-glioma drugs is urgent.

Scutellarin (SCU, 4,5,6-trihydroxyflavone-7-glucuronide) belongs to the active ingredient of flavonoids isolated from Erigeron breviscapus, which holds broad pharmacological effects and has been used to treat cardiac ischemic and cerebral ischemic diseases [8–10]. Among them, scutellarin now has been attracted increasing attention for its antineoplastic effect on most tumors. In colorectal cancer, scutellarin sensitized resveratrol (RSV) and 5-fluorouracil (5-FU)-triggered apoptosis by promoting caspase-6 activation in a P53 dependent manner [11]. In hepatocellular carcinoma and tongue squamous cell carcinoma, scutellarin also could restrain the proliferation and migration of tumor cells [12,13]. However, the role of scutellarin in glioma remains to be further illustrated.

Lidocaine, a derivative of cocaine, is a commonly used local anesthetic and antiarrhythmic drug [14]. In recent years, studies have found that lidocaine exerts anti-tumor effect. Firstly, lidocaine inhibited proliferation of colon cancer cells, and induced their cell-cycle arrest and apoptosis by activation of apoptosis protein pathways [15,16]. What's more, lidocaine suppressed tumor development and enhanced the sensitivity of cisplatin, so combining lidocaine with cisplatin might be a novel treatment for hepatocellular carcinoma and breast cancer [17,18]. Mechanistically, lidocaine and ropivacaine exerted demethylating effects on breast cancer cells at their clinically relevant doses [19]. Furthermore, clinical studies demonstrated that intraoperative intravenous lidocaine infusion was associated with improved overall survival in patients undergoing pancreatectomy [20]. Therefore, lidocaine is a potential and useful antineoplastic agent. Nevertheless, the effect of lidocaine on glioma is still not well-understood.

Epidermal growth factor receptor (EGFR), a versatile signal transducer, plays a critical role in a number of cellular processes such as cellular proliferation, survival, differentiation, migration, inflammation and matrix homeostasis [21]. Frequently, EGFR is elevated in different cancers and this high expression correlates positively with cancer progression and poor prognosis [22]. Glioma, non-small cell lung cancer, head and neck, breast, colorectal, ovarian, prostatic and pancreatic cancers are known to all exhibit increased EGFR activity [23], which is considerably beneficial to survival and growth of these cancer cells [24,25]. In addition, according to

the results predicted by SwissTargetPrediction online tool (http://www.swisstargetprediction.ch/index.php), a web server for target prediction of bioactive small molecules [26], EGFR is the molecular target of scutellarin (S15 Table). Simultaneously, multiple lines of evidence from research suggested that lidocaine inhibited the progression of lung cancer, colorectal cancer and retinoblastoma via regulation of EGFR axis [27–29]. However, whether scutellarin and even its combination with lidocaine exert anti-glioma effect by regulating EGFR signaling remains to be established.

Thus, the aim of this study was to determine the potential antineoplastic effect of scutellarin and its combination with lidocaine on human glioma, and explore the link between the anti-glioma effect and EGFR signaling. The results would provide experimental and theoretical basis for the clinical treatment of glioma.

## Materials and methods

### Cell culture

Glioma cell lines U251 and LN229 were obtained from American Type Culture Collection (ATCC, Manassas, Virginia, USA). Both cell lines were cultured at 37˚C in a humidified incubator with 5% $CO_2$ in Dulbecco's modified Eagle's medium (DMEM) (Hyclone, USA) supplemented with 10% fetal bovine serum (FBS) (HyClone, USA), 100 U/ml penicillin and 100 μg/ml streptomycin (Hyclone, USA). In this study, U251 and LN229 cells with less than 20 passages were used.

For primary culture, astrocytes and neurons were included and used for toxicity analysis. As previously reported [30], astrocytes in the brain of Sprague Dawley (SD) rats were isolated, cultured and purified. In the similar way, cortical neurons were also isolated from neonatal SD rats. All the procedures of animal experiments were conducted with approval by the Ethical Committee of Kunming Medical University (reference number: kmmu 2018016) in this study. After isolation, cortical neurons were resuspended with DMEM medium with 10% FBS and seeded in poly-L-lysine coated cell plates. Four hours (h) later, the DMEM medium was replaced by neuron-specific medium (Neurobasal medium (Gibco, USA): B27 supplement (Gibco, USA) = 50: 1), which was changed every three days. Finally, the cell purity of P2 generation astrocytes and P1 generation cortical neurons cultured for 6 days was evaluated by immunofluorescence staining as described before [31,32].

### Cell viability analysis

According to the instructions, cell viability was analyzed by CCK-8 kit (DOJINDO, Japan). Briefly, 3000~5000 cells were seeded in one well of 96-well plates (Corning, USA). After incubation for 24 h, the drugs (SCU (MedChemExpress, USA; CAS No. 27740-01-8) and lidocaine (aladdin, China; CAS No. L276132)) were administrated. For 48 h of intervention, 10 μl of CCK-8 reagent was added into each well and incubated at 37˚C for 4 h. The absorbance (OD value) was acquired by Multiskan Spectrum Microplate Spectrophotometer (Thermo, USA) at a wavelength of 450 nm.

### Real-time analysis of cell proliferation

The xCELLigence Real Time Cell Analyzer DP Instrument (Roche Diagnostics GmbH, Germany), which could be placed in an incubator containing 5% CO2, 95% humidity and at 37˚C, was applied to continuously monitor the proliferation of U251 cells. Moreover, this apparatus could integrate the relative impedance change of microelectronic sensors on E-plate 16 (Roche Diagnostics GmbH, Germany) bottom, whose output was the cell index used for evaluating

the cell proliferation ability. Briefly, U251 cells were seeded at E-plate 16 with 5000 cells/well. At about 24 h, the drugs were added into the medium. From the time of cell inoculation, the cells were monitored every 15 minutes (min) for a total of 3 days. Data analysis was carried out by xCELLigence Real Time Cell Analyzer software 1.2.

### Plate clone formation assay

As described previously [30], 1000 cells were seeded per well in 6-well plates (Corning, USA). After the cells were attached overnight, the drugs were added into the medium and the cells were cultured in a 37˚C incubator for 14 days. On the 14th day, the cells were fixed with 4% paraformaldehyde for 15 min and then stained with 0.5% crystal violet (Beyotime, China). Finally, a digital camera (Nikon, Japan) was used to take the pictures of the formed clones and the clones were counted. Note: To ensure consistency and reproducibility, the counting of clones was performed by the criteria that one clone had more than 50 cells.

### EdU incorporation assay

EdU incorporation assay were performed by EdU cell proliferation detection kit (RiboBio, China) according to the manufacturer's instructions. In brief, the cells were inoculated into 96-well plates with 5000 cells/well. After 48 h of drug intervention, 100 μl 50 μM EdU medium was added into each well and the plates were incubated at 37˚C for 2 h. Then the cells were fixed for 30 min by 4% paraformaldehyde at room temperature and perforated with 0.5% TritonX-100 solution for 10 min. After rinsing with 0.01 M PBS, the cells were incubated with 1× Apollo® dyeing solution for 30 min at room temperature and in the dark. Finally, the cell nucleus was stained with 1× Hoechst 33342 reaction solution. Images were acquired by an inverted fluorescence microscopy camera system (Leica, Germany). The proliferation rate was equal to the ratio of EdU positive cells (red) to Hoechst positive cells (blue).

### TUNEL assay

The Terminal-deoxynucleoitidyl Transferase Mediated Nick End Labeling (TUNEL) detection kit (Roche, Switzerland) was applied for apoptosis analysis. Following the instructions, the cells were fixed in 4% paraformaldehyde for 15 min at room temperature and then perforated by 0.1% Triton X-100 on ice (2–8˚C) for 2 min. Then the prepared 50 μl TUNEL reaction mixture (TdT: Fluorescein labelled dUTP solution = 1: 9) was added into each well and the reaction was conducted at 37˚C for 1 h in the dark. After rinsing, the cell nuclei were stained with 5 μg/ml DAPI (Beyotime, China). The images were captured by an inverted fluorescence microscopy camera system. The apoptotic rate was equal to the ratio of TUNEL positive cells (green) to DAPI positive cells (blue).

### Apoptosis analysis by flow cytometry

We also conducted cell apoptosis analysis with Annexin-V cell apoptosis detection kit (BD Biosciences, San José, CA) and flow cytometry. As described previously [30], the cells intervened for 48 h were stained with 20 μg/ml Annexin-V labeled with FITC for 30 min and 50 μg/ml PI for 5 min at room temperature and in the dark, respectively. After adding 400 μl of combined buffer, the samples were tested by flow cytometry immediately according to the standards. The living cells were not marked with both Annexin-V and PI, the early apoptotic cells were stained with Annexin-V only, the late apoptotic cells were stained with both Annexin-V and PI, and the mechanically injured cells were only labelled with PI.

## Transwell assay

The migration ability was detected by transwell assay. Briefly, $5\times10^4$ cells intervened by drugs for 48 h were resuspended with serum-free medium and seeded into transwell chambers with matrigel in the bottom (Millipore, Burlington, MA, USA), while the DMEM medium containing 10% FBS was added into the lower chamber. After 48 h of migration, the non-invasive cells in the upper chamber were gently removed, and the invasive cells on the bottom of the upper chamber were fixed with 4% paraformaldehyde and stained with 0.5% crystal violet. After rinsing, the images were photographed by a microscope. Fifteen fields were captured in each group, and the invasive cells were counted and statistically analyzed.

## Tumor xenograft model and therapeutic regimens

All the animal experiments were performed in accordance with the Animal Research: Reporting of *in vivo* Experiments (*ARRIVE*) guidelines in this study. As previously reported [31], female BALB/c nude mice (weighing approximately 20 g and aged 4 to 6 weeks) were included and used to construct a glioma xenograft model by subcutaneously injecting U251 cells ($5 \times 10^6$/mice) into the left side of the armpits of the mice. After the tumors were visible (about 2 weeks after transplantation), the mice were randomly divided into four groups (n = 6 ~ 7): control (10% dimethyl sulfoxide (DMSO)), SCU 50 mg/kg, lidocaine 20 mg/kg, and a combination of SCU (50 mg/kg) and lidocaine (20 mg/kg) (SCU+Lido). Then drug administration was initiated through intraperitoneal injection and lasted for 5 weeks, and the dosing frequency was twice a week. Meanwhile, the tumor volume and body weight of nude mice were examined every 5 days. The tumor volume was calculated as (length × width × width)/2. Finally, the nude mice were euthanized, and the tumors were sampled and weighed.

## Western blotting

Referring to the previous description [30], cells ($5\times10^6$) were lysed for 20 min with lysis buffer (Beyotime, China) containing protease inhibitors (Roche, USA). After centrifugation at 12000×g for 15 min at 4˚C, the protein concentration was determined by BCA method (Beyotime, China). The samples were resolved by SDS/PAGE gel electrophoresis and transferred to PVDF membranes (Immobilon-P membrane, Millipore, Massachusetts, USA). The membranes were blocked with 5% (wt/vol) skimmed milk in TBS for 1 h at room temperature, incubated with primary antibodies-anti-EGFR (Rabbit, 1:1000, Proteintech, China) and β-actin (Mouse, 1:5000, Proteintech) at 4˚C overnight, and then analyzed by immune blotting of HRP-conjugated secondary antibodies (1:5000, GeneTex, USA). An enhanced chemiluminescent (ECL) chromogenic substrate (Biosharp, China) was used to visualize the bands. Blotting was captured by Molecular Imager ChemiDocTM XSR+ Gel Imaging System (Bio-Rad, USA) and analyzed using ImageJ software (NIH). In semi-quantitative analysis of the target protein, each sample was normalized to β-actin.

## Statistical analysis

In this study, the data were expressed as the mean ± standard deviation of the mean (SD). One-way ANOVA followed by LSD or Tamhane's T2 multiple comparisons tests was performed on continuous data from three independent groups and above. Analysis of variance of factorial design was for the factorial design data. All the analysis was performed using SPSS 16.0 software. As long as $P < 0.05$, the difference was statistically significant.

## Results

### 1. Lidocaine inhibited the proliferation of glioma cells in a dose-dependent manner

We detected the cell viability of U251 and LN229 cells intervened with lidocaine for 48 h by CCK8, and found that lidocaine inhibited the proliferation of these two cell lines in a dose-dependent manner (Fig 1A and 1B and S1 Table). Moreover, the IC50 (the concentration when the inhibitory efficiency reaches 50%) of lidocaine in U251 and LN229 cells was 2.531

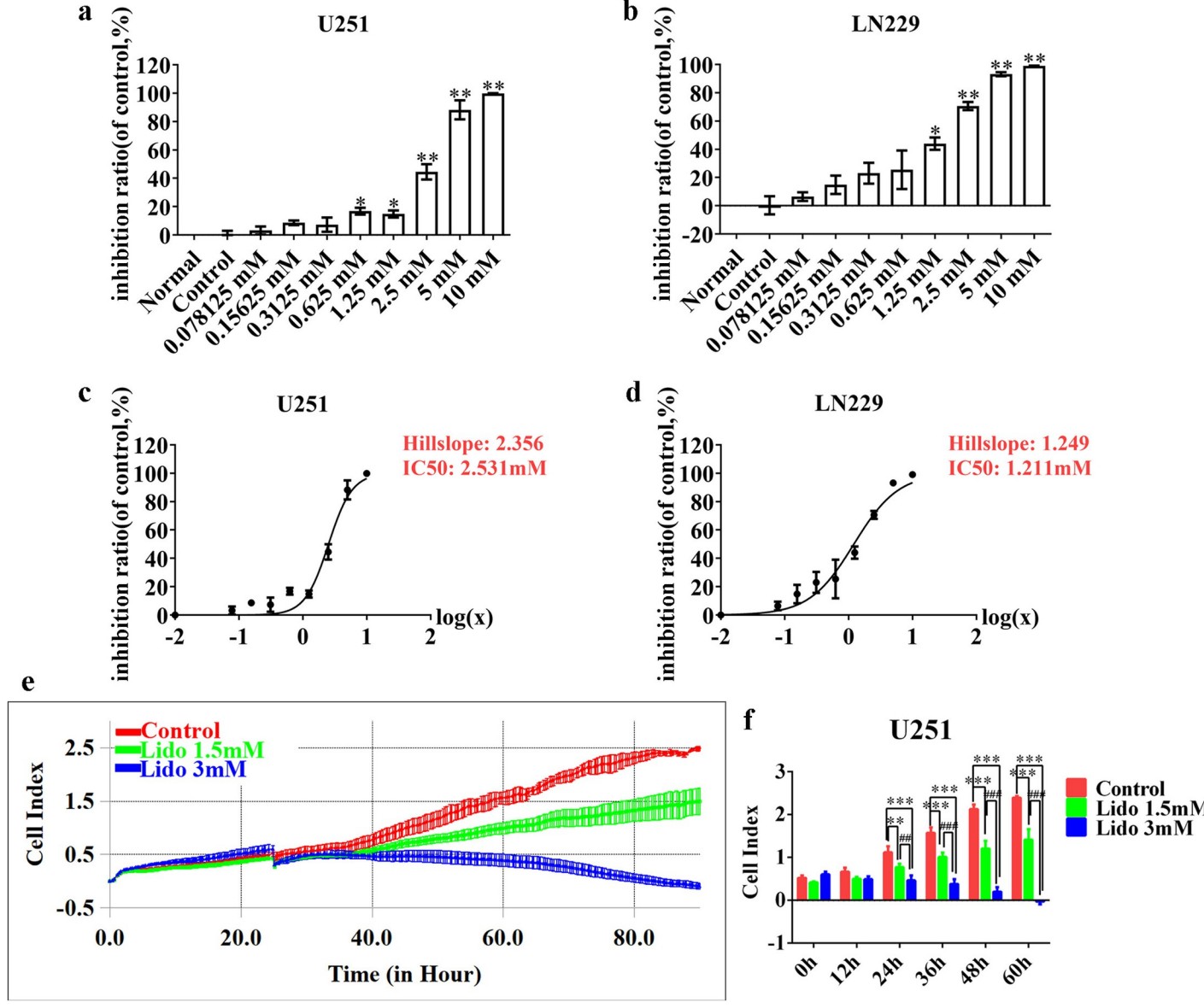

**Fig 1. Effect of lidocaine on the proliferation of glioma cells.** (a, b) the inhibitory rate of different concentrations of lidocaine on U251 and LN229 cells. (c, d) IC50 curve and IC50 of lidocaine in U251 and LN229 cells. (e, f) Lidocaine suppressed the proliferation of U251 cells. e) Cell index was recorded at 15 min interval after inoculation. f) Cell index was recorded after intervened by lidocaine for 0 h, 12 h, 24 h, 36 h, 48 h and 60 h. Normal: Cells without any intervention; Control: Cells with the solvent (5% 0.01M PBS); IC50, the concentration of lidocaine on which the inhibition rate was 50%; Lido: Lidocaine. * vs Control, and # Lido 1.5mM vs Lido 3mM. * P < 0.05, **/## P < 0.01, ***/### P < 0.001 (n = 3).

mM and 1.211 mM, respectively (Fig 1C and 1D). The 95% confidence interval for them was 2.255 mM ~ 2.841 mM and 1.022 mM ~ 1.435 mM, respectively.

In addition, the effect of lidocaine on the proliferation of U251 cells was also continuously monitored by xCELLigence Real Time Cell Analyzer. Our results showed that the cell index curves of proliferation in all groups basically coincided before lidocaine administration. After administration, the cell index of U251 decreased as the dose of lidocaine increased (Fig 1E and 1F and S2 Table). At 24 h of lidocaine intervention, compared with control group (0.133% DMSO), the cell index was going down, and the difference was significant (P < 0.01) (Fig 1E and 1F and S2 Table). With the extension of intervention time, such as at 36 h, 48 h and 60 h, the difference of cell index between lidocaine groups and control group was all statistically significant (P < 0.01) (Fig 1E and 1F and S2 Table). Furthermore, the inhibitory effect of lidocaine 3 mM was better than that of lidocaine 1.5 mM (Fig 1E and 1F and S2 Table).

## 2. Scutellarin and its combination with lidocaine suppressed the proliferation of glioma cells

Many studies have reported that SCU has antineoplastic effect [11–13]. Here we studied the anti-glioma effect of SCU combined with lidocaine. Firstly, the cell viability of U251 and LN229 cells intervened by SCU and its combination with lidocaine 1 mM for 24, 48, and 72 h were detected by CCK8. The results demonstrated that SCU possessed a concentration-dependent inhibitory effect on cell viability of glioma cells (Fig 2A–2C and S3 Table). For U251 cells, it suppressed the cell viability only at high concentration (400 μM) (Fig 2A–2C and S3 Table), but at low concentration (100 or 200 μM) it held a certain proliferation effect, especially at 24 h and 48 h (Fig 2A and 2B and S3 Table). However, low concentration of SCU also showed a non-significant inhibitory effect on cell viability of glioma cells at 72 h (Fig 2C and S3 Table), demonstrating that the inhibitory effect of scutellarin might be also time-dependent. In addition, the proliferation effect of SCU at low concentration could be reversed by lidocaine 1 mM (Fig 2A–2C and S3 Table). For LN229 cells, the inhibition rate was increased with the rise of SCU dose, and the inhibitory effect of SCU on LN229 was enhanced by lidocaine 1 mM (Fig 2A–2C and S3 Table).

Secondly, the cell proliferation was detected by clone formation assay. After the cells adhered to the wall overnight, SCU and its combination with lidocaine 1 mM were administrated for 2 weeks and the medium was changed once every three days (Fig 2D). We found that with the increase of SCU dose, the clone number of U251 and LN229 cells decreased significantly in comparison with control group (P < 0.05) (Fig 2D–2F and S4 Table). Moreover, lidocaine 1 mM also inhibited the clone formation of U251 and LN229 cells (P < 0.05) (Fig 2D–2F and S4 Table). Additionally, the inhibitory effect of SCU was time-dependent. Due to the prolonged SCU exposure (14 days), the inhibitory effect reached the peak at low concentration of SCU (200 μM). Therefore, the addition of lidocaine 1 mM did not show further inhibitory effect on clone formation (Fig 2D–2F and S4 Table). When lidocaine 1 mM was added on the basis of SCU, the promoting effect by lidocaine was only shown at SCU 100 μM (Fig 2D–2F and S4 Table).

Thirdly, the proliferation of LN229 cells intervened with the combination of SCU and lidocaine was also assessed by EdU incorporation assay. According to the results shown in Fig 2A–2C, the inhibitory effect of the combination of SCU and lidocaine peaked at 48 h after dosing, so the proliferation of LN229 cells was detected at this time point. Our results demonstrated that with the rise of SCU dose, the proliferation rate of LN229 cells was significantly reduced in comparison with control group (P < 0.05) (Fig 3A and 3B and S5 Table). Additionally, lidocaine 1 mM also inhibited the proliferation of LN229 cells (P < 0.05) (Fig 3A and 3B and

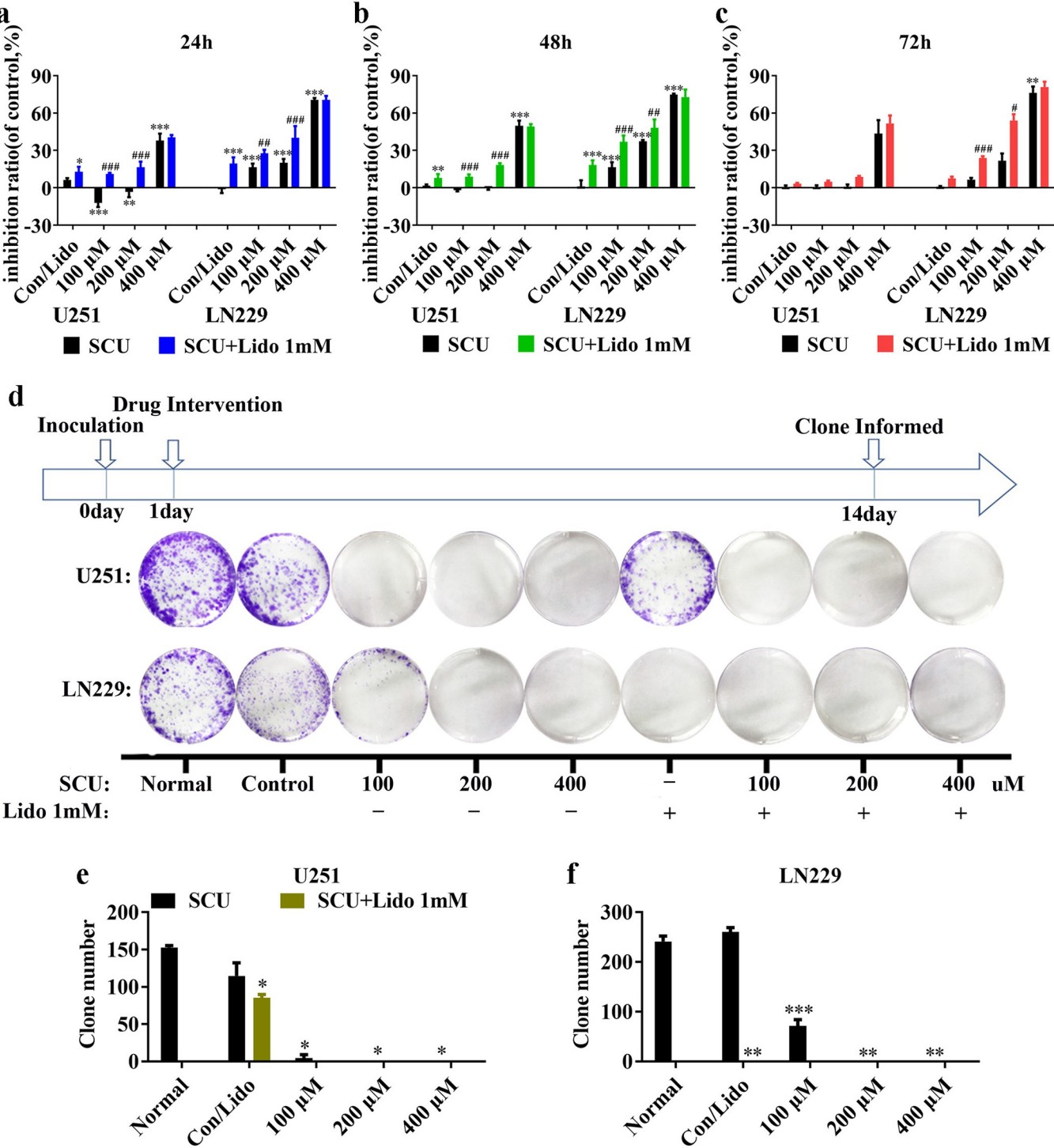

**Fig 2. Scutellarin and its combination with lidocaine inhibited the viability and clone formation of glioma cells.** (a, b, c) Inhibition rate of SCU and its combination with lidocaine 1 mM on U251 and LN229 cells after intervention for 24, 48 and 72 h. (d) The images showed the effect of SCU and its combination with lidocaine on the clone formation of U251 and LN229 cells. (e, f) The clone number of U251 and LN229 cells in (d). The number of cells in one clone was more than 50. SCU: Scutellarin; Lido: Lidocaine; Normal: Cells without any intervention; Con/Control: Cells intervened by 0.133% DMSO. Data are shown as mean ± SD. * vs control group, # SCU x vs SCU x + Lido 1mM. */# P < 0.05, **/## P < 0.01, ***/### P < 0.001 (n = 3).

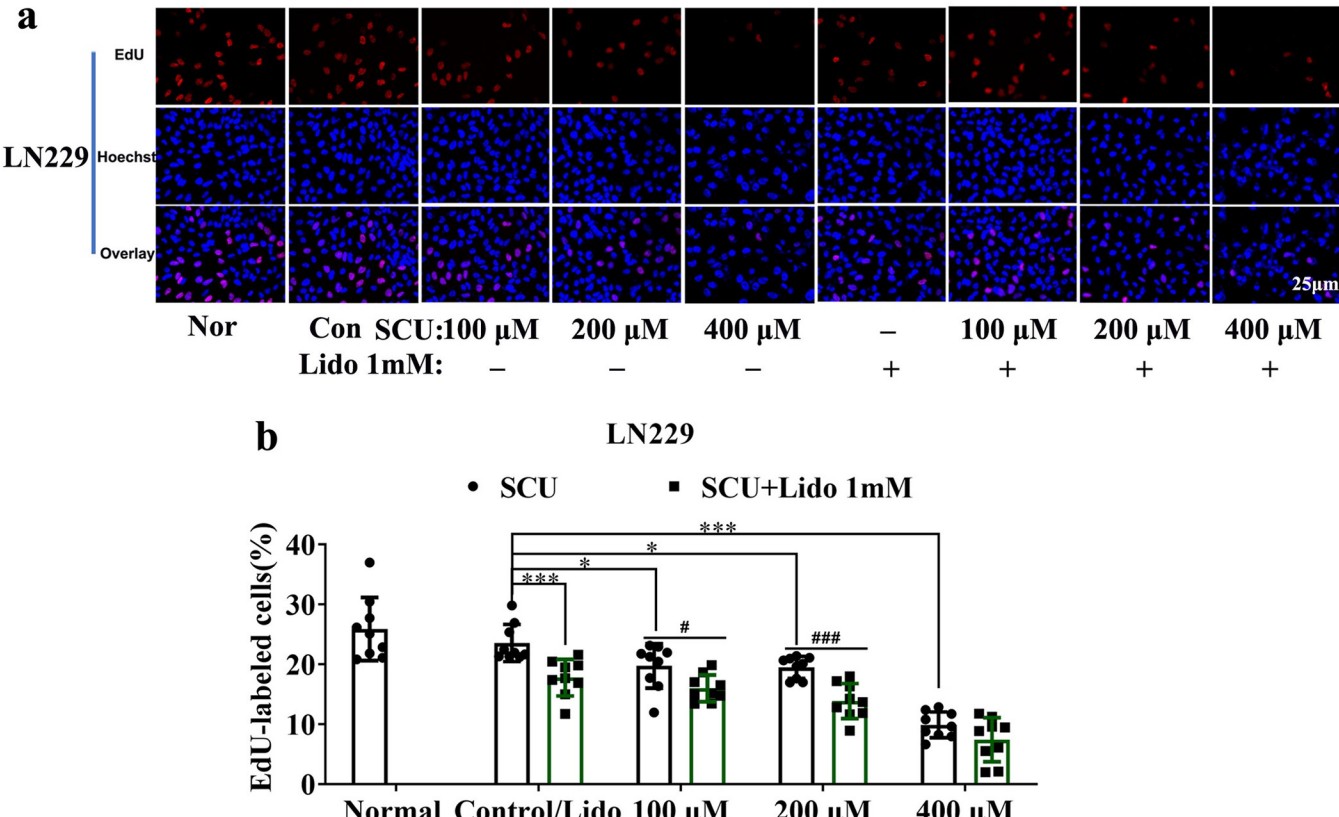

**Fig 3. Scutellarin and its combination with lidocaine suppressed the proliferation of glioma cells.** (a) The effect of SCU and its combination with lidocaine on the proliferation of LN229 cells. (b) Quantitative analysis of the proliferation rate in (a). Nor/Normal: Cells without any intervention; Con/Control: Cells intervened by 0.133% DMSO; SCU: Scutellarin; Lido: Lidocaine. * vs control, # SCU x vs SCU x + Lido 1mM. */# $P < 0.05$, ***/### $P < 0.001$ (n = 9 from 3 replicate wells).

S5 Table). Moreover, the combination of SCU and lidocaine 1 mM could further inhibit the proliferation of LN229 cells (Fig 3A and 3B and S5 Table).

## 3. Scutellarin and its combination with lidocaine induced the apoptosis of glioma cells

After intervened by SCU and its combination with lidocaine for 48 h, the apoptosis of U251 and LN229 cells was evaluated by both TUNEL staining and flow cytometry. In TUNEL assay, we found that SCU promoted the apoptosis of U251 and LN229 cells in a concentration-dependent manner (Fig 4A–4C and S6 Table). Moreover, compared with control group, the apoptosis rate of U251 and LN229 cells were also increased by lidocaine 1 mM (Fig 4A–4C and S6 Table), and lidocaine 1 mM could further improve the effect of inducing apoptosis by SCU (Fig 4A–4C and S6 Table).

Additionally, the result of flow cytometry was consistent with that of TUNEL assay. With the increase of SCU dose, the early apoptosis rate and late apoptosis rate (i.e. necrosis) of U251 and LN229 cells were increased after 48 h of intervention (Fig 4D–4F, S7 Table and S1 and S2 Files). Moreover, lidocaine 1.5 mM raised the early and late apoptosis rate of U251 and LN229 cells compared with control group (Fig 4D–4F, S7 Table, and S1 and S2 Files). Furthermore, the early and late apoptosis rate of U251 and LN229 cells intervened by the combination of

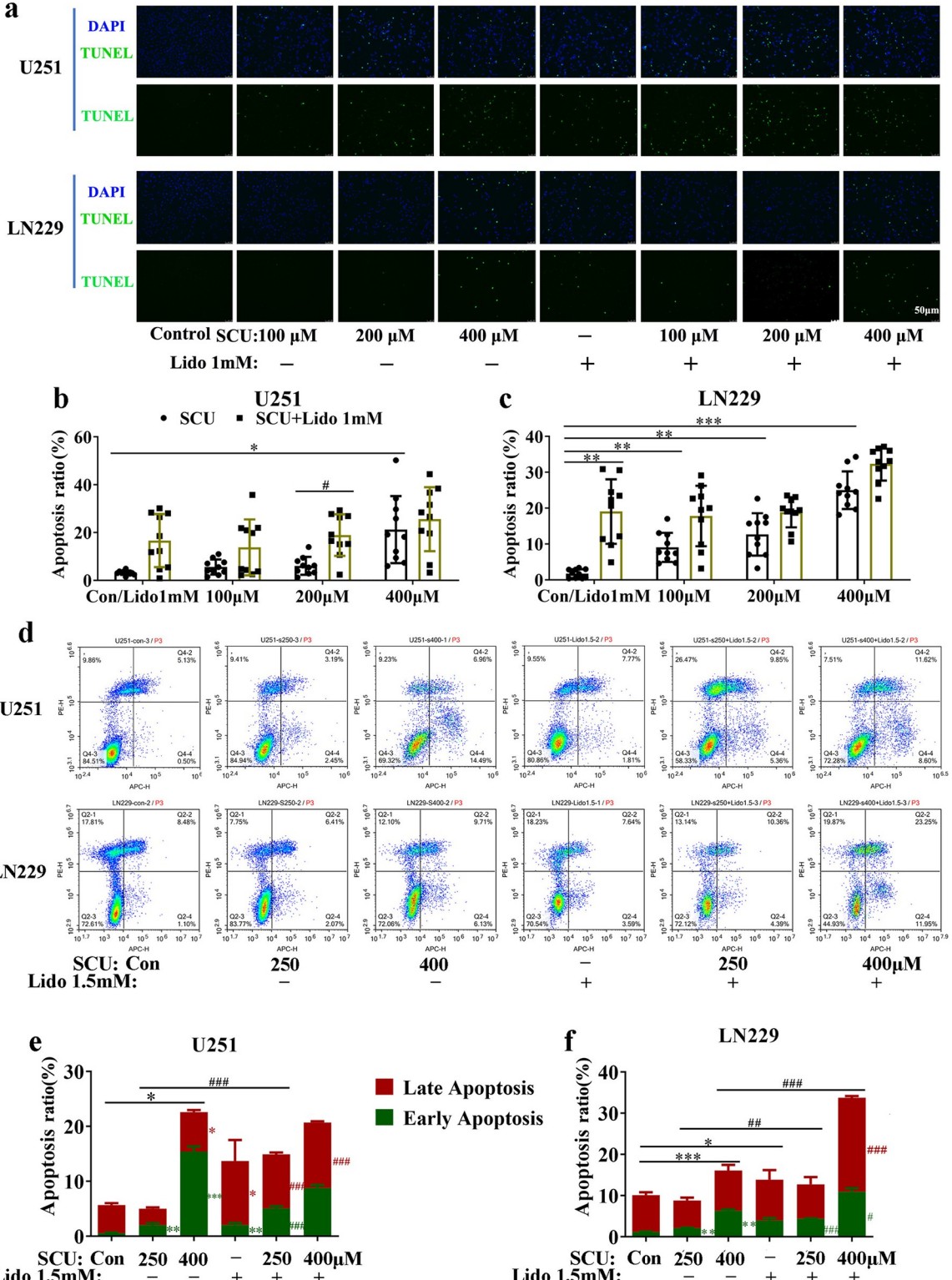

**Fig 4. Scutellarin and its combination with lidocaine induced the apoptosis of glioma cells.** (a)Fluorescence images by TUNEL assay showed the apoptosis of U251 and LN229 cells induced by SCU and its combination with lidocaine 1 mM. (b, c) Quantification of apoptosis rate of U251 and LN229 cells induced by SCU and its combination with lidocaine 1 mM (n = 10 from 3 replicate samples). (d) The apoptosis diagrams of U251 and LN229 cells induced by SCU and its combination with lidocaine 1.5 mM. (e, f) Quantification of the early, late and total apoptosis rate of U251 and LN229 cells by flow cytometry (n = 3). Early apoptosis: See the second quadrant (Q4-2) in each single figure of Fig 4D; Late apoptosis: See the fourth quadrant (Q4-4) in each single figure of Fig 4D; Total apoptosis: Sum of

early apoptosis and late apoptosis. Con/Control: Cells intervened by 0.133% DMSO; SCU: Scutellarin; Lido: Lidocaine. * vs Control, # SCU x vs SCU x + Lido 1/1.5mM. */# P < 0.05, **/## P < 0.01, ***/### P < 0.001. */# (black) for apoptosis rate in (b, c) or total apoptosis rate in (e, f); */# (green) for early apoptosis rate in (e, f); */# (red) for late apoptosis rate in (e, f).

SCU and lidocaine 1.5 mM was higher than that intervened with SCU or lidocaine alone (P < 0.05) (Fig 4D–4F, S7 Table and S1 and S2 Files).

## 4. Scutellarin and its combination with lidocaine repressed the migration of glioma cells

The migration ability of U251 and LN229 cells intervened by SCU and its combination with lidocaine for 48 h was determined by transwell assay. The results indicated that with the increase of SCU dose, the number of U251 and LN229 cells migrating to the bottom of the chambers was significantly declined (P < 0.05) compared with control group (Fig 5A–5C and S8 Table). Lidocaine 1 mM also inhibited the migration of U251 cells to the bottom of the chambers (P < 0.05) (Fig 5A and 5B and S8 Table). Furthermore, the combination of SCU with lidocaine 1 mM could further repress the migration of U251 and LN229 cells (Fig 5A–5C and S8 Table). The difference between SCU alone groups and the combination groups was statistically significant (P < 0.05) (Fig 5B and 5C).

## 5. Scutellarin and its combination with lidocaine also suppressed the growth of glioma *in vivo*

To further validate the antitumor characteristics of SCU and lidocaine, we established an *in vivo* xenograft tumor model of glioma and intraperitoneally injected SCU 50 mg/kg, lidocaine

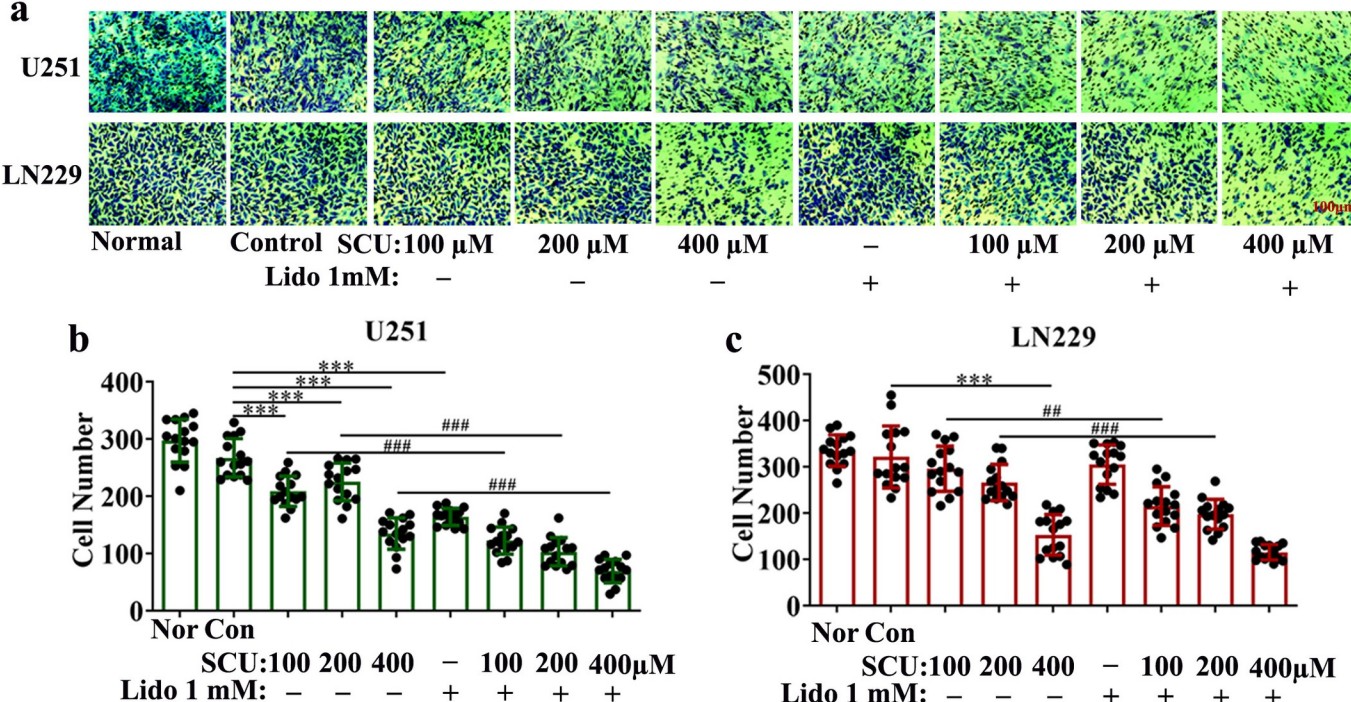

**Fig 5. Scutellarin and its combination with lidocaine suppressed the migration of glioma cells.** (a)Images showed the U251 and LN229 cells that vertically migrated to the bottom of the chambers after intervention by SCU and its combination with lidocaine 1 mM for 48 h. (b, c) Quantification of the U251 and LN229 cells that vertically migrated to the bottom of the chambers. Nor/Normal: Cells without any intervention; Con/Control: Cells intervened by 0.133% DMSO; SCU: Scutellarin; Lido: Lidocaine. * vs Control, # SCU x vs SCU x + Lido 1 mM. ## P < 0.01, ***/### P < 0.001 (n = 15 from 3 replicate samples).

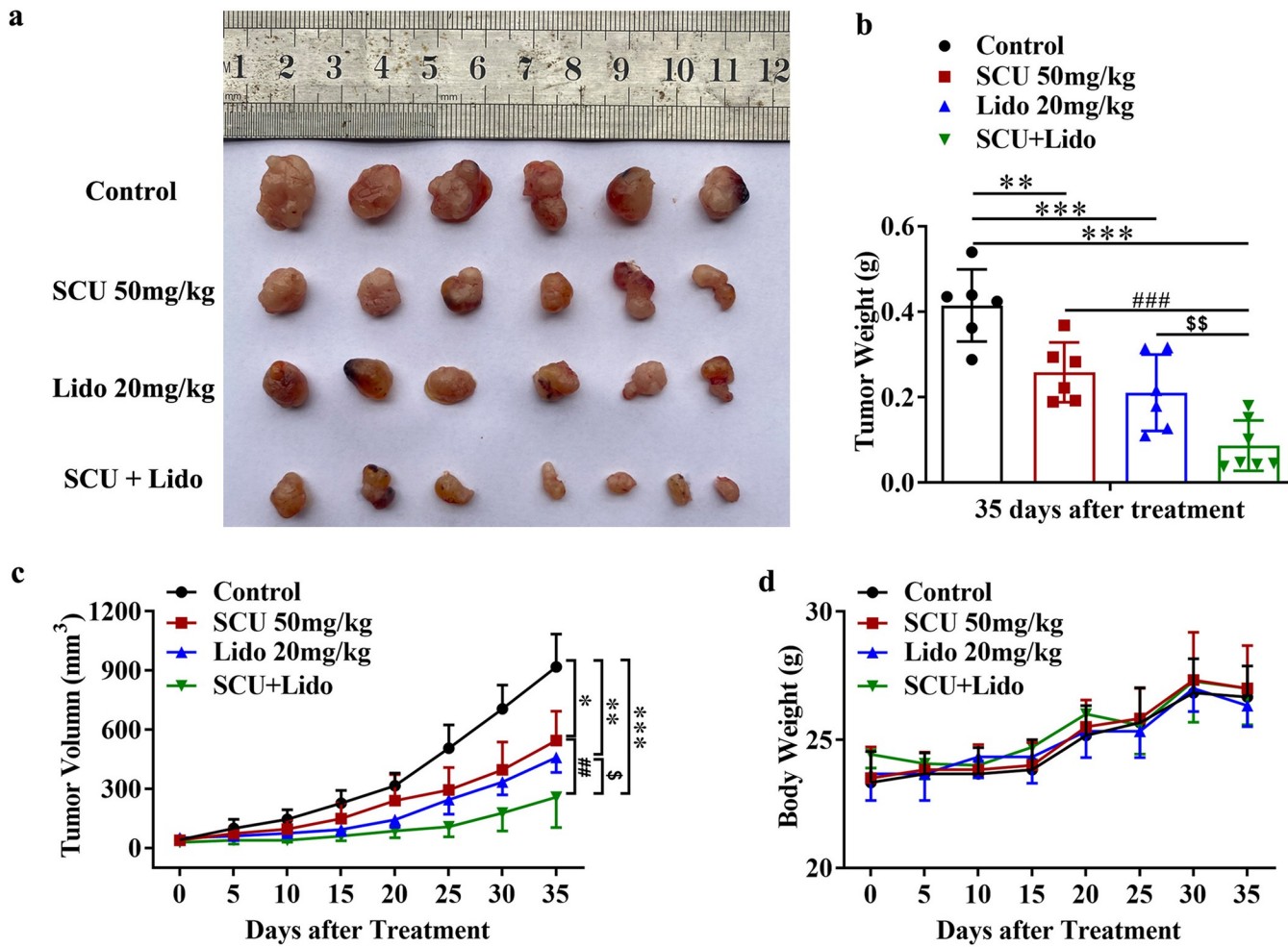

**Fig 6. Scutellarin and its combination with lidocaine also suppressed the growth of glioma *in vivo*.** (a, b) The obtained xenografts (a) and tumor weight (b) after administrated with SCU and its combination with lidocaine for 35 days. (c) The tumor volume after administrated with SCU and its combination with lidocaine. (d) The body weight of nude mice after administrated with SCU and its combination with lidocaine. SCU: Scutellarin; Lido: Lidocaine. * vs Control, # SCU vs SCU + Lido, $ Lido vs SCU + Lido. */$ P < 0.05, **/##/$ $ P < 0.01, ***/### P < 0.001 (n = 6 ~ 7).

20 mg/kg, and the combination (SCU 50 mg/kg and lidocaine 20 mg/kg) into nude mice to observe the growth of tumor. Our results demonstrated that the tumor size, weight and volume in SCU 50 mg/kg and lidocaine 20 mg/kg groups were significantly smaller than those in control group (10% DMSO) (Fig 6A–6C and S9 Table), suggesting that SCU and lidocaine with the above doses could effectively inhibit tumor growth *in vivo*. Moreover, the combination achieved a better inhibitory effect on tumor growth (Fig 6A–6C and S9 Table). However, the body weight of nude mice was not obviously different among these four groups (Fig 6D and S9 Table), indicating that SCU 50 mg/kg, lidocaine 20 mg/kg and even the combination held no effect on the growth of mice.

## 6. The toxicity of scutellarin and its combination with lidocaine on astrocytes and neurons

In this study, astrocytes and cortical neurons from newborn SD rats were isolated, cultured and used for the toxicity analysis of SCU and its combination with lidocaine. Here, the cell

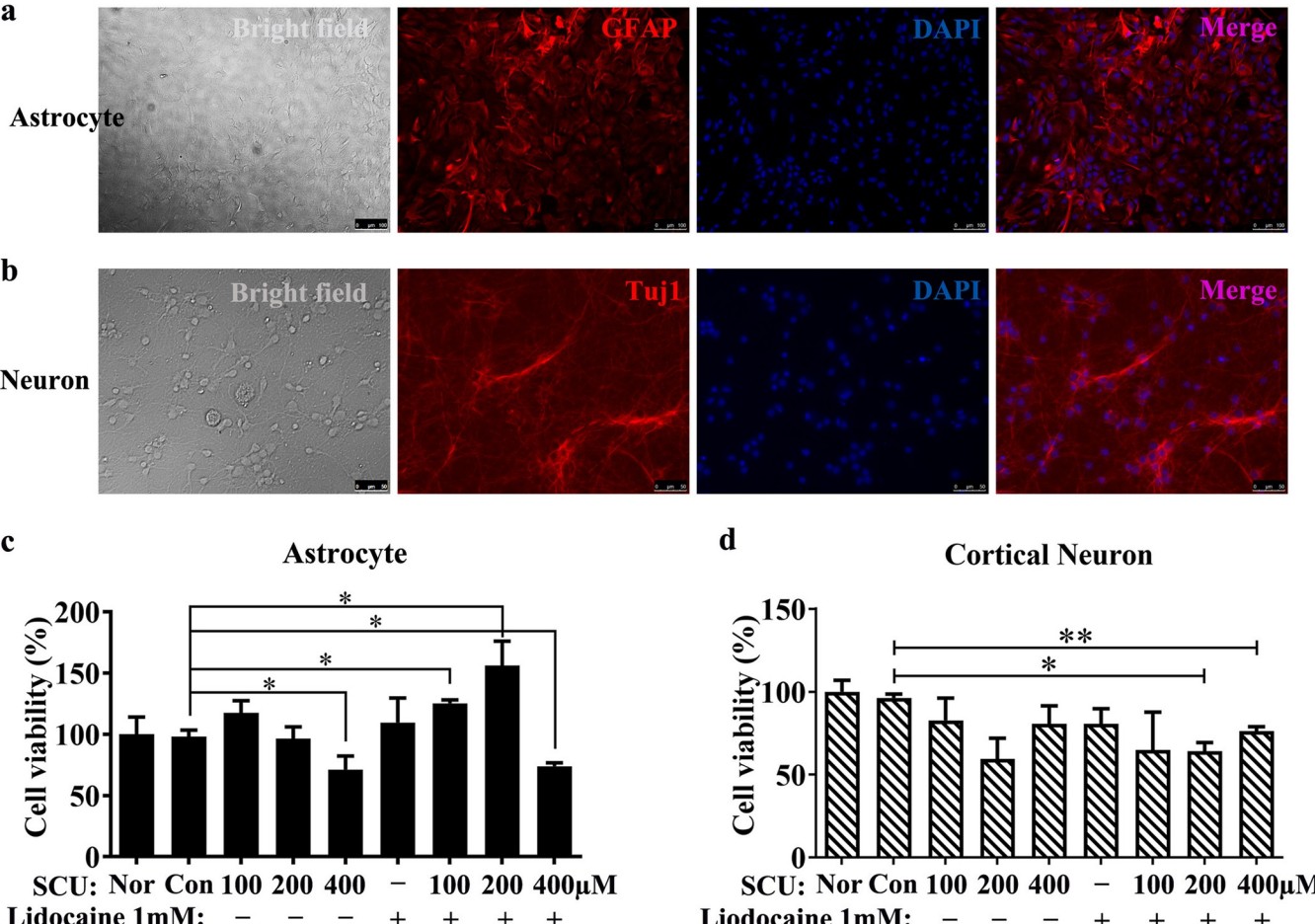

**Fig 7. The toxicity of scutellarin and its combination with lidocaine on astrocytes and neurons.** (a, b) The purity of P2 generation astrocytes and P1 generation cortical neurons cultured for 6 days in this study. (c, d) The cell viability of astrocytes (c) and cortical neurons (d) after administrated with SCU and its combination with lidocaine 1 mM for 48 h. GFAP: Marker for astrocytes, Tuj1: Marker for neurons, Nor: Normal cells without any intervention, con: Control group, the cells were intervened with 0.133% DMSO, SCU: Scutellarin. * compared to control group, * P < 0.05, ** P < 0.01 (n = 3).

purity of astrocytes (P2 generation) and cortical neurons (P1 generation) reached 90% or more (Fig 7A and 7B). After 48 h of intervention with SCU and its combination with lidocaine 1 mM, the cell viability of these two cells was determined by CCK8 kit.

For astrocyte, we found that as the dose of SCU increased, its cell viability declined (Fig 7C and S10 Table). Moreover, SCU 400 μM significantly attenuated the cell viability of astrocytes compared to control group (Fig 7C and S10 Table, P < 0.05). In addition, lidocaine 1 mM harbored no effect on cell viability of astrocytes (Fig 7C and S10 Table). When SCU was combined with lidocaine 1 mM, the cell viability of astrocytes in SCU 100 μM plus lidocaine 1 mM group and SCU 200 μM plus lidocaine 1 mM group was even higher than that in control group (Fig 7C and S10 Table, P < 0.05). However, the cell viability in SCU 400 μM plus lidocaine 1 mM group was significantly lower than that in control group, but similar to that in SCU 400 μM group (Fig 7C and S10 Table). The above results indicated that only high dose of SCU (like 400 μM) was toxic to astrocytes, while lidocaine 1 mM was non-toxic to astrocytes and also did not aggravate the toxicity of SCU.

For cortical neuron, although its cell viability slightly decreased in SCU 200 μM group and SCU 400 μM group compared to control group, there was no statistical difference (Fig 7D and

S10 Table, P > 0.05), revealing that SCU held low toxicity to cortical neurons. In addition, lidocaine 1 mM showed no toxic effect on the cell viability of cortical neurons (Fig 7D and S10 Table). However, compared with control group, the cell viability of cortical neurons in SCU 200 μM plus lidocaine 1 mM group and SCU 400 μM plus lidocaine 1 mM group was significantly reduced (Fig 7D and S10 Table, P < 0.05). Interestingly, the cell viability of cortical neurons in the combination of SCU and lidocaine 1 mM groups was not different from that in SCU alone groups (Fig 7D and S10 Table), suggesting that lidocaine 1 mM did not intensify the toxicity of SCU on cortical neurons.

## 7. EGFR signaling was repressed by scutellarin and its combination with lidocaine

In order to explore the possible molecular mechanism of the anti-glioma effect of SCU combined with lidocaine, we applied malacards (the human disease database, https://www.malacards.org/) to find the genes related to glioblastoma (see S11 Table). The results revealed that EGFR was related to glioblastoma with the highest score after PTEN (see S11 Table). Further analysis demonstrated that EGFR was involved in many biological processes and pathways related to glioblastoma according to GeneCards Suite gene sharing, such as negative regulation of apoptotic process, positive regulation of ERK1 and ERK2 cascade, EGFR signaling pathway, ERBB2 signaling pathway, cellular response to drug and MAPK cascade in the top 10 biological processes and almost all the top 20 pathways (Fig 8A and 8B and S11 Table). These results indicated that EGFR signaling was closely related to the occurrence and development of glioma. Furthermore, through the UALCAN database (http://ualcan.path.uab.edu/index.html), we found that the mRNA expression of EGFR was up-regulated in primary glioblastomas in comparison with normal tissues, and the difference was statistically significant (P = 1.13220000041991E-08) (Fig 8C). Functionally, EGFR activator (100 μM; CAS No. NSC228155, TOPSCIENCE, China) increased the cell viability and the formed clones, but induced the apoptosis of U251 and LN229 cells (Fig 8D–8K and S12–S14 Tables). However, the EGFR inhibitor (20 nM; CAS No. 879127-07-8, TOPSCIENCE, China) held the opposite effects (Fig 8D–8K and S12–S14 Tables).

Simultaneously, according to the predicted results by SwissTargetPrediction online tool (shown in S15 Table), EGFR was the molecular target of SCU. Moreover, multiple lines of evidence from research suggested that lidocaine inhibited the progression of lung cancer, colorectal cancer and retinoblastoma via regulating EGFR axis [27–29]. Therefore, we speculated that SCU and even its combination with lidocaine might exert the anti-glioma effect by regulating EGFR signaling. Interestingly and dramatically, the results of western blot demonstrated that SCU downregulated the EGFR protein level of U251 and LN229 cells in a dose-dependent manner, and the difference between SCU groups and control group was statistically significant (P < 0.05) (Figs 8L and 8M and S1). Moreover, the EGFR protein expression of U251 and LN229 cells in lidocaine 1.5 mM group was also declined (P < 0.05) (Figs 8L and 8M and S1). More importantly, the EGFR protein level of U251 and LN229 cells intervened by the combination of SCU and lidocaine 1.5 mM was less than that intervened with SCU or lidocaine alone (P < 0.05) (Figs 8L and 8M and S1).

## Discussion

In this study, scutellarin combined with lidocaine was applied as a new therapeutic strategy for human glioma. We found that single-agent SCU and lidocaine suppressed the proliferation and migration, and induced the apoptosis of glioma cells in a dose-dependent manner *in vitro*. When administered with the combination of SCU and lidocaine, the proliferation and

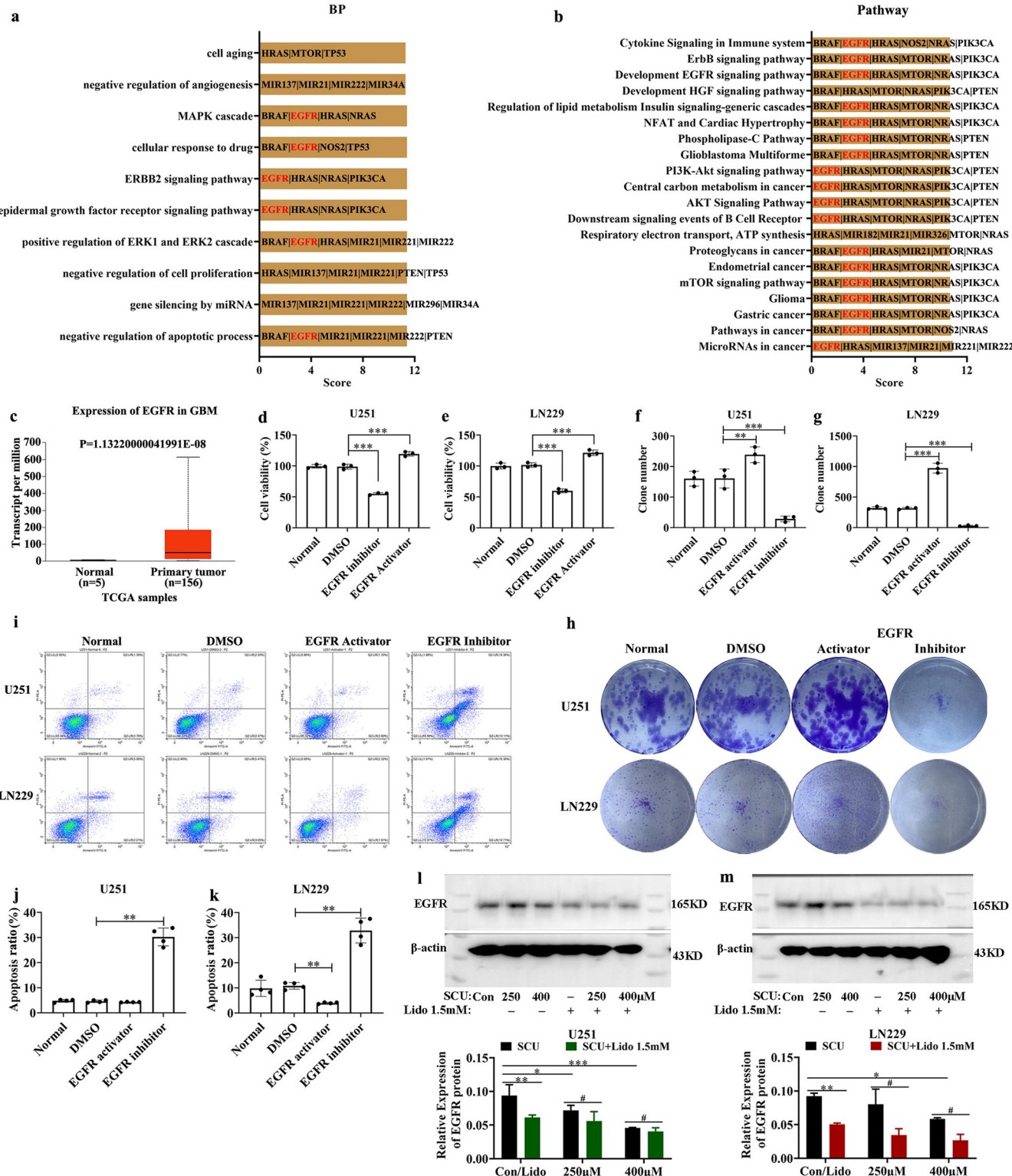

**Fig 8. The EGFR expression in glioma cells intervened by SCU and its combination of lidocaine was downregulated.** (a) The top 10 biological processes related to glioblastoma according to GeneCards Suite gene sharing. (b) The top 20 pathways related to glioblastoma according to GeneCards Suite gene sharing. (c) The mRNA expression of EGFR in normal tissue and primary glioblastoma multiforme (GBM). (d, e) The cell viability of U251 and LN229 cells after intervened by EGFR activator (100 µM) and inhibitor (20 nM) for 24 h (n = 3). (f-h) The formed clones of U251 and LN229 cells after intervened by EGFR activator and inhibitor for 14 days (n = 3). (f, g) the clone number, (h) the representative pictures. (i-k) The apoptosis of U251 and LN229 cells after

intervened by EGFR activator and inhibitor for 24 h. (i) the representative pictures, (j-k) the apoptosis rate (n = 4). (l, m) The relative protein level of EGFR in U251 and LN229 cells intervened by SCU and its combination of lidocaine for 48 h (n = 3). The bloting bands from three repeats were shown in S1 Fig. Normal: Cells without any intervention in (d-k); DMSO: Cells intervened by 0.3% DMSO in (d-k); Con: Cells intervened by 0.133% DMSO; SCU: Scutellarin; Lido: Lidocaine. */# P < 0.05, ** P < 0.01, *** P < 0.001.

migration ability of glioma cells were further reduced, and the apoptosis was also higher *in vitro*, that is, the two drugs exerted a synergistic effect (Fig 9). Additionally, the combination of these two drugs also synergistically inhibited the growth of glioma *in vivo*. Most importantly, the toxicity of SCU and its combination with lidocaine was low to normal astrocytes and neurons. Mechanistically, the antineoplastic effect of SCU and its combination with lidocaine on glioma was partially associated with the repression of EGFR signaling (Fig 9). These results could provide a reference for the treatment of glioma, and the combination of scutellarin and lidocaine might become a new chemotherapy method in future.

Scutellarin, an extractant of the Chinese herbal medicine Erigeron breviscapus, exerts the anti-tumor effect in many types of tumors, including colorectal cancer, tongue squamous carcinoma, hepatocellular carcinoma, prostate cancer, etc [11,13,33–35]. The previous studies have shown that scutellarin at low dose (10 μM) could induce cell cycle arrest at G0/G1 transition by downregulating the expression of cyclin D1 and CDK4, and at high dose (15 μM or higher) could induce apoptosis by promoting caspases activation [36]. For example, Liu X and his colleagues found scutellarin could regulate the cell cycle of cancer cells and induce apoptosis in lymphohematological tumors [37]. In addition, scutellarin held a strong PKM2 activation effect and could inhibit cell growth [38,39]. Consistent with the previous studies, we also found that scutellarin suppressed the proliferation/growth and migration, and induced the apoptosis of glioma cells *in vitro* and *in vivo* in present study.

Many studies have reported that lidocaine also held anti-cancer effect. In clinically relevant concentrations, lidocaine harbored the significant antiproliferative effect on human

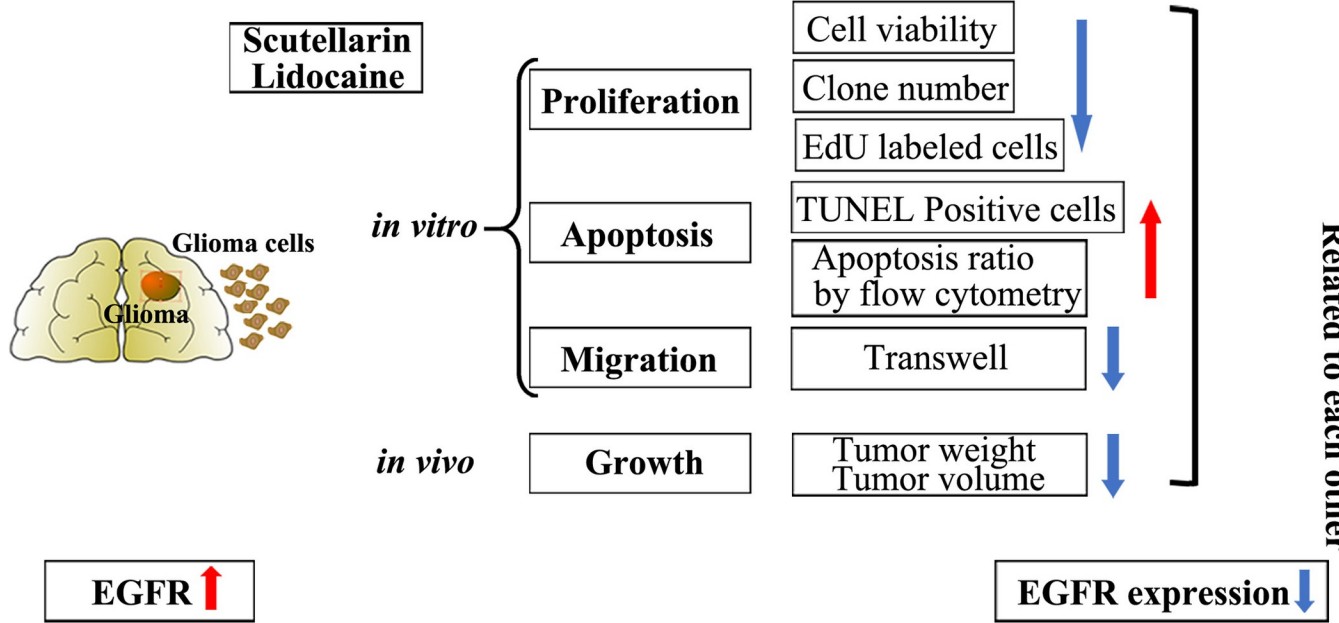

**Fig 9. A graphical abstract depicting the antineoplastic effect of scutellarin combined with lidocaine on human glioma.** Scutellarin combined with lidocaine suppressed the proliferation/growth and migration, and induced the apoptosis of glioma cells *in vitro* and *in vivo*. In addition, the EGFR expression was up-regulated in glioma. However, EGFR expression was reversed by scutellarin combined with lidocaine in glioma cells.

hepatocarcinoma cells by modifying the P53 expression level, and these effects were time and dose-dependent [40]. In addition, lidocaine inhibited the viability and migration of breast cancer cells [41]. Moreover, intraperitoneal lidocaine improved survival of mice with MDA-MB-231 peritoneal carcinomatosis, in which the lidocaine dose was consistent with the current clinical setting for analgesia [41]. Although the anti-tumor effect of lidocaine was not the first discovery in this study, it was first used for human glioma. Similar to previous results, we found that lidocaine suppressed the proliferation/growth and migration, and induced the apoptosis of glioma cells *in vitro* and *in vivo*. In agreement with the notion that local anesthesia might be beneficial to cancer therapy.

As is known to all, the combination of multi-drugs at low-dose could not only reduce drug resistance and improve effectiveness compared to monotherapy, but also reduce the toxicity of high-dose single drug [42,43]. Therefore, we studied the combined effect of scutellarin and lidocaine on glioma. We found that these two drugs held synergistical anti-glioma effect. Actually, studies have reported that SCU and lidocaine not only possessed anti-tumor effect, but also could improve the sensitivity of cancer cells to other chemotherapeutic drugs. For example, SCU could be used as a sensitizer for cisplatin and 5-FU in several tumors. In ovarian cancer, scutellarin formed a complex with cisplatin to cause a greater conformational change in DNA, which finally induced DNA strand breaks and enhanced apoptotic signaling pathway response [44]. Moreover, scutellarin was capable of sensitizing non-small cell lung cancer and prostate cancer cells to cisplatin by enhancing caspases-dependent apoptosis and cytotoxic autophagy [45,46]. Additionally, the administration of scutellarin sensitized 5-FU-evoked apoptosis in p53[+] colon cancer cells and prompted 5-FU-elicited caspase-6 activation in a time-dependent manner [11,43]. Despite being a classic regional anesthetic, lidocaine also enhanced the sensitivity of cisplatin to tumor cells. In hepatocellular carcinoma and breast cancer, combining lidocaine with cisplatin could further suppress tumor development and metastasis compared to cisplatin monochemotherapy [17,18]. Therefore, co-administration of lidocaine during cisplatin chemotherapy seemed warranted in the clinical treatment of these tumors. In addition to well-known therapeutic agents (cisplatin and 5- FU), a combination with novel anti-tumor agents also induced more potent therapeutic responses [43]. For example, the combination of scutellarin and $C_{18}H_{17}NO_6$ (caffeoyl tyrosine) further inhibited glioma proliferation/growth and migration and promoted apoptosis compared to single-drug administration [31]. Based on the above evidence, it's not difficult to understand that scutellarin and lidocaine were well-matched with each other to result in more strong anti-glioma responses and reduce the toxicity to normal cells.

EGFR is a transmembrane glycoprotein and a member of the tyrosine kinase superfamily receptor [47]. Many EGFR gene alterations have been identified in gliomas, including amplifications, deletions and single nucleotide polymorphisms (SNPs) [48–50], so it has served as a clinical biomarker in gliomas [48,50]. Simultaneously, EGFR was the molecular target of SCU predicted by SwissTargetPrediction online tool. Moreover, lidocaine suppressed the viability, migration and invasion of lung cancer cells, but induced apoptotic death by blocking EGFR signaling [27]. Further, lidocaine reduced the proliferation and induced the apoptosis of retinoblastoma cells by decreasing EGFR expression [29]. Based on this and the results from malacards database (https://www.malacards.org/) (see S11 Table), we detected the EGFR protein level in glioma cells intervened by scutellarin and its combination with lidocaine. Our results demonstrated that scutellarin and lidocaine also separately downregulated the EGFR protein level of glioma cells in dose-dependent manner. Furthermore, the combination of SCU and lidocaine could further decline the EGFR protein expression. EGFR is known to regulate several downstream signaling pathways such as ERK, PI3K/AKT and JAK/STAT, which all play a crucial role in the regulation of cancer stemness, angiogenesis and metastasis [22]. It has been

proven that scutellarin and lidocaine could exert anti-tumor effects by inhibiting these downstream signaling pathways. Firstly, scutellarin induced cell cycle arrest, apoptosis and autophagy, and also inhibited migration and invasion of lung cancer and human leukemia cells through suppressing ERK, AKT and STAT3 signaling pathways [51–53]. Secondly, lidocaine also blocked tumor progression and recurrence by influencing the activation of ERK and PI3K/AKT signaling pathways [27,54,55]. Based on the above, we believed that the repression of EGFR signaling might be the potential mechanism that scutellarin combined with lidocaine exerted the anti-gliome effect.

In conclusion, scutellarin combined with lidocaine suppressed the proliferation and migration and induced the apoptosis of glioma cells, in which the underlying mechanism might be partly associated with the repression in EGFR signaling (Fig 9). These results prompted the therapeutic prospect of these two drugs.

## Supporting information

**S1 Fig. The original western blot for three repeats.**
(TIF)

**S1 Table. The raw data for inhibition ratio of lidocaine on glioma cells by CCK8.**
(XLS)

**S2 Table. The raw data for proliferation of glioma cells intervened with lidocaine by xCELLigence Real Time Cell Analyzer.**
(XLS)

**S3 Table. The raw data for inhibition ratio of scutellarin and its combination with lidocaine on glioma cells by CCK8.**
(XLS)

**S4 Table. The raw data for clone number of glioma cells intervened by scutellarin and its combination with lidocaine.**
(XLSX)

**S5 Table. The raw data for proliferation rate of LN229 cells by EdU incorporation assay.**
(XLSX)

**S6 Table. The raw data for apoptosis rate of glioma cells by TUNEL assay.**
(XLSX)

**S7 Table. The raw data for apoptosis rate of glioma cells by flow cytometry analysis.**
(XLSX)

**S8 Table. The raw data for transferred rate of glioma cells by transwell assay.**
(XLSX)

**S9 Table. The raw data for tumor weight, tumor volume and body weight of nude mice *in vivo* experiment.**
(XLSX)

**S10 Table. The raw data for the toxicity analysis of scutellarin and its combination with lidocaine on astrocytes and cortical neurons.**
(XLSX)

**S11 Table. The genes related with glioblastoma by Malacards.**
(XLSX)

**S12 Table. The raw data for cell viability of glioma cells after intervened by EGFR activator and inhibitor.**
(XLSX)

**S13 Table. The raw data for clone number of glioma cells after EGFR activator and inhibitor intervention.**
(XLSX)

**S14 Table. The raw data for apoptosis of glioma cells induced by EGFR activator and inhibitor.**
(XLSX)

**S15 Table. The molecular targets of scutellarin and lidocaine predicted by SwissTargetPrediction.**
(XLSX)

**S1 File. The report for apoptosis analysis of U251 cells by flow cytometry.**
(PDF)

**S2 File. The report for apoptosis analysis of LN229 cells by flow cytometry.**
(PDF)

## Acknowledgments

We would like to thank Xiao-Jiao Wang (Core Facilities of West China Hospital, Sichuan University) for the apoptosis analysis by flow cytometry.

## Author Contributions

**Conceptualization:** Xiu-Ying He, Qing-Jie Xia, Xiao-Ming Zhao, Ting-Hua Wang.

**Data curation:** Yue-Xiang Zheng.

**Formal analysis:** Xiu-Ying He, Yui-Si Yang, Yue-Xiang Zheng, Qing-Jie Xia, Hong-Zhou Yu.

**Funding acquisition:** Xiu-Ying He, Xiao-Ming Zhao.

**Investigation:** Yui-Si Yang, Xiao-Ming Zhao, Ting-Hua Wang.

**Methodology:** Xiu-Ying He, Yui-Si Yang, Yue-Xiang Zheng, Qing-Jie Xia, Hong-Zhou Yu, Ting-Hua Wang.

**Project administration:** Xiu-Ying He, Ting-Hua Wang.

**Supervision:** Qing-Jie Xia, Xiao-Ming Zhao, Ting-Hua Wang.

**Validation:** Xiu-Ying He.

**Writing – original draft:** Xiu-Ying He.

**Writing – review & editing:** Xiu-Ying He, Ting-Hua Wang.

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
