## [Decision Letter · Decision Letter 0]

20 May 2024

PONE-D-23-32383Scutellarin combined with lidocaine attenuates growth of human glioma cells associated with down-regulation of epidermal growth factor receptorPLOS ONE

Dear Dr. He,

Thank you for submitting your manuscript to PLOS ONE. After careful consideration, we feel that it has merit but does not fully meet PLOS ONE’s publication criteria as it currently stands. Therefore, we invite you to submit a revised version of the manuscript that addresses the points raised during the review process.

We look forward to receiving your revised manuscript.

Kind regards,

Jianhong Zhou

Staff Editor

PLOS ONE

Journal Requirements:

   "This work was supported by Natural Science Foundation of Sichuan Province of China (Grant No. 2023NSFSC1567) and Research fund projects of Yunnan Education Department (Grant No. 2020J0066)."

Reviewers' comments:

Reviewer's Responses to Questions

**Comments to the Author**

1. Is the manuscript technically sound, and do the data support the conclusions?

Reviewer #1: Partly

Reviewer #2: Yes

Reviewer #3: Yes

2. Has the statistical analysis been performed appropriately and rigorously? 

Reviewer #1: N/A

Reviewer #2: Yes

Reviewer #3: Yes

3. Have the authors made all data underlying the findings in their manuscript fully available?

Reviewer #1: Yes

Reviewer #2: Yes

Reviewer #3: Yes

4. Is the manuscript presented in an intelligible fashion and written in standard English?

Reviewer #1: Yes

Reviewer #2: Yes

Reviewer #3: Yes

5. Review Comments to the Author

Reviewer #1: In this manuscript, the authors tell a story about the effects of Scutellarin-lidocaine on human glioma cells. The results of this work indicated that Scutellarin-lidocaine could repress the proliferation and promote the apoptosis of U251 and LN229 cells. However, there are some problems.

1. Scutellarin and lidocaine have been widely reported in diseases, including various tumors. The authors should cite more references and make more discussions to reveal why such a combination was designed.

2. Where is SCU from (the manufacturer and product No.), how about its purity? BP? AP? SP? or UP? Would the impurities in the sample affect the cells?

3. It's not strange that SCU and lidocaine could affect the proliferation, migration and apoptosis of glioma cells. In many other reports, both SCU and lidocaine could also affect the properties of other tumor cells and normal cells, e.g., in our previous work, SCU and lidocaine could affect vascular endothelial cells, Müller cells and fibroblast cells via regulating STAT3 associated axes. Therefore, my suggestion is to use some normal cells as blank, to investigate their differences with the same drug doses.

4. Actually, EdU assay is to investigate the DNA synthesis but not proliferation, and PCNA could be used as a proliferation biomarker.

5. Fig 2 seems confusing, in some cases, low concentration of SCU could even promote the proliferation of U251, and at 400 uM, the addition of lidocaine did not promote the effects of SCU. Therefore, a normal blank cell is necessary here.

6. In Figure 4d, the authors should carefully check the apoptosis figures, which reveal similar contours. After scanning by some software, several ones seem like to be the same one at different time.

7. The most important problem of this work is the absent of mechanism. The authors indicated that Scutellarin-lidocaine could downregulate EGFR, however, what's the direct target of SCU and what's the target of lidocaine? The section between Scutellarin-lidocaine and EGFR is still a black box. Logically, the authors should at least predict the target of SCU via some online tools, e.g., SwissTargetPrediction, screen some ones and confirm them by Biacore, nano-ITC, IP or in vivo navigation.

In the recent stage, the manuscript is not an integrated story.

Reviewer #2: Scutellarin combined with lidocaine attenuates growth of human glioma cells

associated with down-regulation of epidermal growth factor receptor

Comments

In the current article, Xiu et al., had found that a combination of Scutellarin and lidocaine attenuates growth of human glioma cells by down-regulation of epidermal growth factor receptor. Overall, the study has several strengths, including the use of multiple assays to assess various aspects of glioma cell behaviour, such as proliferation, migration, and apoptosis. Additionally, investigating the combination of two compounds, scutellarin and lidocaine, adds novelty to the research. However, there are some considerations and areas for improvement in the study design and future research:

Major Suggestions:

• Can you clarify the rationale behind the choice of cell lines used in the study?

• While the general cell culture conditions are provided, it would be beneficial to mention the passage number of the cells used. Continuous passaging can lead to genetic drift and altered cellular behaviour, affecting experimental reproducibility.

• The choice of CCK-8 assay for cell viability analysis is common and suitable. However, the duration of drug intervention (48 hours) seems arbitrary and lacks justification. Providing a rationale for this time frame would enhance the clarity and reproducibility of the experiment.

• I recommend that the authors conduct cytotoxicity analysis on normal astrocytes to confirm the toxic impact of the two compounds.

• Additionally, the effect of these drugs should be compared with the standard drug for glioma viz., Temozolomide through in vitro assays to validate their efficacy.

• For the plate clone formation assay, it would be helpful to include information on the criteria used for defining a clone (e.g., minimum number of cells required) and how the counting of clones was performed to ensure consistency and reproducibility.

• Considering the heterogeneity of Glioma, incorporating 3D culture models such as spheroid or organoid cultures would enhance the relevance and translatability of the findings to in vivo settings.

• The study proposes a combination therapy approach of Scutellarin and lidocaine against Glioma. Have similar combination therapies been explored in clinical trials or preclinical studies for GBM, and what are the challenges associated with translating this approach to clinical settings?

• While the study implicates the downregulation of EGFR protein as a potential mechanism for the observed effects, additional experiments has to be conducted to further elucidate the molecular pathways involved. In that case, which downstream signalling cascades will the authors study and why?

Minor Suggestions:

• Check for grammatical accuracy and maintain a consistent writing style.

Overall, the paper has a solid foundation, but refining and expanding on certain aspects will contribute to a more comprehensive and reader-friendly manuscript.

Reviewer #3: He et al attempted to explore the role of Scutellarin and lidocaine in glioma growth. Both compound inhibited glioma proliferation. They found interesting results. This study open the window for glioma therapeutic approach.

6. PLOS authors have the option to publish the peer review history of their article (what does this mean?). If published, this will include your full peer review and any attached files.

Reviewer #1: **Yes: **Jun Shao

Reviewer #2: **Yes: **Daisy Precilla S

Reviewer #3: **Yes: **Mehdi Hayat Shahi

---

## [Author Response · Author response to Decision Letter 0]

31 Jul 2024

The response to reviewers' comments is shown in ‘Response to Reviewers’ file.

---

## [Decision Letter · Decision Letter 1]

10 Sep 2024

PONE-D-23-32383R1Scutellarin combined with lidocaine exerts antineoplastic effect in human glioma associated with repression of epidermal growth factor receptor signalingPLOS ONE

Dear Dr. He,

Thank you for submitting your manuscript to PLOS ONE. After careful consideration, we feel that it has merit but does not fully meet PLOS ONE’s publication criteria as it currently stands. Therefore, we invite you to submit a revised version of the manuscript that addresses the points raised by one of the reviewers during the review process. Specifically, check your figures, provide data on purity of scutellarin and clarify your manuscript on minor concerns.  Please submit your revised manuscript by Oct 25 2024 11:59PM. If you will need more time than this to complete your revisions, please reply to this message or contact the journal office at plosone@plos.org. Please include the following items when submitting your revised manuscript:A rebuttal letter that responds to each point raised by the academic editor and reviewer(s). You should upload this letter as a separate file labeled 'Response to Reviewers'.A marked-up copy of your manuscript that highlights changes made to the original version. You should upload this as a separate file labeled 'Revised Manuscript with Track Changes'.An unmarked version of your revised paper without tracked changes. You should upload this as a separate file labeled 'Manuscript'.If applicable, we recommend that you deposit your laboratory protocols in protocols.io to enhance the reproducibility of your results. Protocols.io assigns your protocol its own identifier (DOI) so that it can be cited independently in the future. For instructions see: https://journals.plos.org/plosone/s/submission-guidelines#loc-laboratory-protocols. Additionally, PLOS ONE offers an option for publishing peer-reviewed Lab Protocol articles, which describe protocols hosted on protocols.io. Read more information on sharing protocols at https://plos.org/protocols?utm_medium=editorial-email&utm_source=authorletters&utm_campaign=protocols.

We look forward to receiving your revised manuscript.

Kind regards,

Surinder K. Batra

Academic Editor

PLOS ONE

Journal Requirements:

Reviewers' comments:

Reviewer's Responses to Questions

**Comments to the Author**

1. If the authors have adequately addressed your comments raised in a previous round of review and you feel that this manuscript is now acceptable for publication, you may indicate that here to bypass the “Comments to the Author” section, enter your conflict of interest statement in the “Confidential to Editor” section, and submit your "Accept" recommendation.

Reviewer #1: All comments have been addressed

Reviewer #3: All comments have been addressed

2. Is the manuscript technically sound, and do the data support the conclusions?

Reviewer #1: Yes

Reviewer #3: Yes

3. Has the statistical analysis been performed appropriately and rigorously? 

Reviewer #1: Yes

Reviewer #3: Yes

4. Have the authors made all data underlying the findings in their manuscript fully available?

Reviewer #1: Yes

Reviewer #3: Yes

5. Is the manuscript presented in an intelligible fashion and written in standard English?

Reviewer #1: Yes

Reviewer #3: Yes

6. Review Comments to the Author

Reviewer #1: 1. CCK8 assay is only used to test the DNA synthesis or duplication, but not cell proliferation, it should be clarified in the whole manuscript.

2. In the authors' response, "scutellarin (CAS No. 27740-01-8) was purchased from MedChemExpress Company

(MCE). Its purity has reached 98.56%, which could eliminate the influence of the impurities on cells." Could a 98.56% purity eliminate the influence of the impurities on cells?

3. Could the authors carefully check the apoptosis figures, several sub-images showed the similar contours.

Reviewer #3: Combination of Scutellarin and Lidocaine as antineoplastic potential drug for glioma. This is significance study and add new knowledge in glioma tumour therapeutic approach.

7. PLOS authors have the option to publish the peer review history of their article (what does this mean?). If published, this will include your full peer review and any attached files.

Reviewer #1: No

Reviewer #3: **Yes: **Mehdi Hayat Shahi

---

## [Author Response · Author response to Decision Letter 1]

14 Sep 2024

Journal Requirements:

Answer: We have reviewed the reference list and made corrections. Now we ensure that the reference list is complete and correct. And no retracted articles were cited in this manuscript.

Reviewers' Comments:

Reviewer #1: 

1. CCK8 assay is only used to test the DNA synthesis or duplication, but not cell proliferation, it should be clarified in the whole manuscript.

Answer: Dear reviewer, many thanks for your comment.

According to the directions, Cell Counting Kit-8 (CCK8) utilizes highly sensitive water-soluble tetrazolium salt-WST-8 as a chromogenic substrate, which can be reduced by intracellular dehydrogenases to produce water-soluble formazan dye. The number of live cells is directly proportional to the amount of formazan dye. Based on this, this kit is usually used for cell counting when detecting the effects of chemical drugs on cell proliferation or toxicity. Therefore, we believe that it is reasonable to use this kit to detect the effect of scutellarin and lidocaine on cell proliferation of glioma cells in this study. In fact, cell proliferation is accompanied by DNA synthesis and replication, so the results of CCK8 assay can also reflect the synthesis and replication of DNA in cells.

2. In the authors' response, "scutellarin (CAS No. 27740-01-8) was purchased from MedChemExpress Company (MCE). Its purity has reached 98.56%, which could eliminate the influence of the impurities on cells." Could a 98.56% purity eliminate the influence of the impurities on cells?

Answer: Dear reviewer, thank you for pointing this out.

The description ‘a 98.56% purity eliminate the influence of the impurities on cells’ is incorrect. We apologize for misleading you and the readers with such a description. What we intended to express is that the purity of scutellarin (CAS No. 27740-01-8) is very high, which can reduce the influence of impurities on cells. However, whether or to what extent the impurities make an impact on cells requires further investigation.

3. Could the authors carefully check the apoptosis figures, several sub-images showed the similar contours.

Answer: Dear reviewer, we humbly acknowledge your comment. 

We agree with you that the observed contours of apoptosis figures from the same cell line in Fig 4d appear similar. However, these figures are portraying different observations from the different interventions experimented in our study. For instance, the figures in the upper row of Fig 4d reflect the apoptosis of U251 cells with different drug interventions, and the figures in the bottom row of Fig 4d show the apoptosis of LN229 cells with different drug interventions. The interventions involved the use of scutellarin drug at different concentrations with or without a 1.5mM lidocaine mixture. These interventions were specifically as follows: the concentration of scutellarin at (i) 250μM without lidocaine, (ii) 400μM without lidocaine, (iii) 250μM with lidocaine, (iv) 400μM with lidocaine, and lastly (v) the cells with only 1.5mM lidocaine. Based on these aforementioned interventions, we can clearly see that both early and late apoptosis in each of the cell lines U251 and LN229, are statistically different. As shown in Fig 4e-f, scutellarin promoted early apoptosis (see the second quadrant (Q4-2) in each single figure of Fig 4d) and late apoptosis (see the fourth quadrant (Q4-4) in each single figure of Fig 4d) of U251 and LN229 cells in a concentration-dependent manner. Lidocaine 1.5 mM also raised the early and late apoptosis of U251 and LN229 cells compared with control group (Fig 4. e, f). What's more, the early and late apoptosis rate of U251 and LN229 cells intervened by the combination of scutellarin and lidocaine 1.5mM was higher than that intervened with scutellarin or lidocaine alone (Fig 4. e, f). We are therefore confident that the figures in Fig 4d are different from each other as they show results of samples taken from different interventions. 

The reasons why the observed contours of apoptosis figures appear similar may be as follows: 1) the samples are sourced from the same cell line. As shown in Fig 4d, the figures from the same row appear similar, but the figures from the different row (the upper row: U251 cells; the bottom row: LN229 cells) are obviously different. 2) During the implementation of the experiment, all conditions were controlled uniformly, which reduced the interference of other factors on the experimental results and improved the reproducibility of the findings.

Reviewer #3: Combination of Scutellarin and Lidocaine as antineoplastic potential drug for glioma. This is significance study and add new knowledge in glioma tumor therapeutic approach.

Answer: Dear reviewer, many thanks for your comments.

Note: All the revisions have been marked with red fonts in the revised manuscript.

---

## [Decision Letter · Decision Letter 2]

8 Jan 2025

Scutellarin combined with lidocaine exerts antineoplastic effect in human glioma associated with repression of epidermal growth factor receptor signaling

PONE-D-23-32383R2

Dear Dr. He,

We’re pleased to inform you that your manuscript has been judged scientifically suitable for publication and will be formally accepted for publication once it meets all outstanding technical requirements.

Kind regards,

Rashmi Rana, PhD

Academic Editor

PLOS ONE

Additional Editor Comments (optional):

Reviewers' comments:

Reviewer's Responses to Questions

**Comments to the Author**

1. If the authors have adequately addressed your comments raised in a previous round of review and you feel that this manuscript is now acceptable for publication, you may indicate that here to bypass the “Comments to the Author” section, enter your conflict of interest statement in the “Confidential to Editor” section, and submit your "Accept" recommendation.

Reviewer #1: (No Response)

Reviewer #3: All comments have been addressed

Reviewer #4: All comments have been addressed

2. Is the manuscript technically sound, and do the data support the conclusions?

Reviewer #1: Partly

Reviewer #3: Yes

Reviewer #4: Yes

3. Has the statistical analysis been performed appropriately and rigorously? 

Reviewer #1: N/A

Reviewer #3: Yes

Reviewer #4: Yes

4. Have the authors made all data underlying the findings in their manuscript fully available?

Reviewer #1: (No Response)

Reviewer #3: Yes

Reviewer #4: Yes

5. Is the manuscript presented in an intelligible fashion and written in standard English?

Reviewer #1: (No Response)

Reviewer #3: Yes

Reviewer #4: Yes

6. Review Comments to the Author

Reviewer #1: 1. Usually, to investigate the cell proliferation, simply CCK-8 assay is not enough, and it should be used together with other protocols, e.g. Edu, PCNA or MTT assays.

2. Usually, for pharmaceutical tests, the compounds should be further purified via RP-HPLC, TLC or other protocols if conditions permit. Or the impurities should be carefully characterized.

Reviewer #3: (No Response)

Reviewer #4: Please change the figure 1 and 2 Y-axis labeling should be changed from inhibite ratio to inhibition ratio.

7. PLOS authors have the option to publish the peer review history of their article (what does this mean?). If published, this will include your full peer review and any attached files.

Reviewer #1: No

Reviewer #3: **Yes: **MEHDI HAYAT SHAHI

Reviewer #4: No

---

## [Editor Report · Acceptance letter]

23 Jan 2025

PONE-D-23-32383R2 

PLOS ONE

Dear Dr. He, 

I'm pleased to inform you that your manuscript has been deemed suitable for publication in PLOS ONE. Congratulations! Your manuscript is now being handed over to our production team.

Kind regards, 

on behalf of

Dr. Rashmi Rana 

Academic Editor

PLOS ONE